# Experimental Evaluation of the Thermoelectrical Performance of Photovoltaic-Thermal Systems with a Water-Cooled Heat Sink

**Husam Abdulrasool Hasan** [1,*], **Jenan S. Sherza** [1], **Jasim M. Mahdi** [2,*], **Hussein Togun** [3,4], **Azher M. Abed** [5], **Raed Khalid Ibrahim** [6] and **Wahiba Yaïci** [7]

1    Department of Air Conditioning & Refrigeration Techniques, Al-Esra'a University College,
     Baghdad 10068, Iraq
2    Department of Energy Engineering, University of Baghdad, Baghdad 10071, Iraq
3    Department of Biomedical Engineering, University of Thi-Qar, Nassiriya 64001, Iraq
4    College of Engineering, University of Warith Al-Anbiyaa, Karbala 56001, Iraq
5    Air Conditioning and Refrigeration Techniques Engineering Department, Al-Mustaqbal University College,
     Babylon 51001, Iraq
6    Department of Medical Instrumentation Engineering, Al-Farahidi University, Baghdad 10015, Iraq
7    CanmetENERGY Research Centre, Natural Resources Canada, 1 Haanel Drive, Ottawa, ON K1A 1M1, Canada
*    Correspondence: drhusam@esraa.edu.iq (H.A.H.); jasim@siu.edu (J.M.M.)

**Abstract:** A design for a photovoltaic-thermal (PVT) assembly with a water-cooled heat sink was planned, constructed, and experimentally evaluated in the climatic conditions of the southern region of Iraq during the summertime. The water-cooled heat sink was applied to thermally manage the PV cells, in order to boost the electrical output of the PVT system. A set of temperature sensors was installed to monitor the water intake, exit, and cell temperatures. The climatic parameters including the wind velocity, atmospheric pressure, and solar irradiation were also monitored on a daily basis. The effects of solar irradiation on the average PV temperature, electrical power, and overall electrical-thermal efficiency were investigated. The findings indicate that the PV temperature would increase from 65 to 73 °C, when the solar irradiation increases from 500 to 960 W/m$^2$, with and without cooling, respectively. Meanwhile, the output power increased from 35 to 55 W when the solar irradiation increased from 500 to 960 W/m$^2$ during the daytime. The impact of varying the mass flow rate of cooling water in the range of 4 to 16 L/min was also examined, and it was found that the cell temperature declines as the water flow increases in intensity throughout the daytime. The maximum cell temperature recorded for PV modules without cooling was in the middle of the day. The lowest cell temperature was also recorded in the middle of the day for a PVT solar system with 16 L/min of cooling water.

**Keywords:** photovoltaic-thermal; PVT solar system; water-cooled heat sink; heat transfer; electrical efficiency; thermal efficiency

## 1. Introduction

The extremely high demands on resources such as water, electricity, housing, and energy as a result of the tremendous increase in population is a major challenge that almost all the world suffers from. Due to these demands, there has been a significant rise in the use of fossil fuels. It is anticipated that the worldwide demand for energy will increase by a factor of 50% between the years 2015 to 2040, as a direct result of the rapid increase in population and the subsequent growth in economic activity. This increase leads to more dependence on fossil fuels, which produce harsh levels of greenhouse gases and air pollution, both of which contribute to the occurrence of natural disasters such as scarcities of water resources, windstorms, and photochemical haze. There is also an interest worldwide to transition toward renewable energy alternatives such as wind, solar, and geothermal

energy. Solar energy has the potential to be seen as a clean kind of energy that is always available with no cost to use. Solar collectors can harness the sun's heat, while photovoltaic cells can convert sunlight into electricity. PV cells are one of the technologies that have been the most popular for the clean and ecofriendly production of electricity in the last decades. Since solar cells are often installed outdoors, they are vulnerable to the varying weather conditions that might have an effect, either directly or indirectly, on their level of output and efficiency. Numerous studies have shown that the ambient air temperature is the factor that has the ultimate bearing on the conversion efficiencies of solar cells. The efficacy of the devices, which should ideally be between 15% and 20%, is affected by a fall of up to 0.4–0.5% for every 1 degree over 25 °C, which is the standard temperature of operation.

Several research studies have been undertaken to lower the temperature of PV modules using air and water as cooling fluids. The systems that combine the cooling techniques with PV modules are termed hybrid photovoltaic–thermal systems (PVT). Effective application of PVT modules offers the advantage of producing both electricity and thermal power at the same time [1]. According to Kharchenko et al. [2], these modules are mainly categorized as the planar liquid type and the concentrator type. The planar type, which is the main theme of this study, provides a simpler design and a substantial cost reduction. Meanwhile, the other type is highly complex and causes difficulties in the manufacturing process, thus leading to higher costs [2]. Shen et al. [3] developed a set of parallel cooling channels to study the impact of water-based cooling on the overall performance of a PVT collector. The collector was found to became more evenly distributed in temperature when the main channel and subchannel diameter ratios increased, so that the channels with D/d = 4 could be 50% greater than those with D/d = 2. López-Álvarez et al. [4] developed an operation-control algorithm for hybrid concentrated solar power (CSP) with solar PVs in the south of Spain and reported good retrofitting possibilities for the CSP plants in operation. Kim et al. [5] showed that the building-integrated photovoltaic–thermal (BIPVT) system could produce 30 W more power than the BIPV system, and the maximum difference in temperature between BIPVT and BIPV was about 22 °C during the winter season. Hussain and Kim [6] used a dual-fluid heat exchanger to cool the PV cells in which water and air are operated simultaneously. The study showed that the annual electrical and total thermal efficiencies with a glass-to-glass configuration are 14.3% and 52.2% compared to 13.9% and 48.3% for the glass-to-PV back-sheet configuration, respectively. Noro and Lazzarin [7] tested what would happen if evacuated tube collectors were put together with a PVT system to power a heat-pump-based plant that works as a chiller in the summer. Pokorny and Matuška [8] tested the effect of a glazed PVT collector for domestic hot water preparation in multifamily buildings. The results showed that a PVT system installed on the roof has a higher thermal yield than a conventional system, with an increase in the thermal yield from 37% to 53%. Gomaa et al. [9] assessed the performance of a concentrating PVT system to produce electric and thermal power under different operating conditions and found that the maximum electrical and thermal energy obtained were 170 and 580 kW. Husam et al. [10] used (water-$Al_2O_3$) nanofluid as a cooling fluid to improve the electrical efficiency of a Fresnel-based Concentrated Photovoltaic (CPVt) system and discovered that the nanofluid used can provide a 23% heat transfer enhancement over water alone. The top surface temperature decreases as the Nusselt number increases.

Ould-Lahoucine et al. [11] used water-$TiO_2$ nanofluid as coolant for a PVT collector to determine its energetic and exegetic properties and found that increasing heat transfer had a significant impact on total energy efficiency, exegetic efficiencies, thermal efficiencies, and electrical efficiencies. Shittu et al. [12] used a micro/channel heat pipe to enhance the electrical conversion efficiency for a PVT collector utilizing a wider solar spectrum, for better energy harvesting from the inward solar radiation. The obtained data showed that the micro/channel heat pipe is a feasible cooling technique to improve the electrical performance of hybrid PVT systems. Varmira et al. [13] studied the effect of the flow of water-glycerol/silver nanofluid in a sheet–sinusoidal tube on the energetic and exegetic behavior in the PVT collector. The effect of various mass flow rates and volume concentra-

tions of silver of nanofluid were examined experimentally, and the results showed better thermofluidic performance at higher mass flow rates and concentrations of nanofluid. Yu et al. [14] used parallel cooling channels with expanded grooves to improve the energy conversion of a PVT collector and found that the mean PV temperature with groove channels was about four degrees cooler compared with smooth channels. Ul Abdin and Rachid [15] showed that the increase in wind speed led to a drop in the thermal efficiency of PVT systems from 70% to approximately 40%. The result also presents that the rise in the thickness of the insulating layers results in a quicker variation in the outlet PVT temperature. Xiao et al. [16] studied the heat-transfer enhancement, thermal efficiency, and temperatures of components for stepped photovoltaic–thermal solar-collector with a solar still. The cooling channel was considered under the base of the solar still, marking to enhance heat-transfer rates. Freshwater productivity was investigated in order to calculate the performance of the solar-still system. The results showed that the overall heat transfer coefficients between the absorber plate and saline water were enhanced by 44%, thereby, the mean temperature of saline water was raised by 16% and the daily freshwater productivity was improved by 52%. Hissouf et al. [17] investigated the improved efficiency of a PVT collector by employing three different geometrical channel-shapes for circulating the coolant fluids. Pure water and an ethylene glycol–water (EG–Water) mixture were used as cooling fluids. The results presented that the half-tube channel design offers the highest efficiency for a photovoltaic–thermal solar collector. Thermal efficiency improved by 1.17%, with the flow cooling the fluid of $m^o = 0.04$ kg/s inside the half tube, and 2.6%, compared to square and circular tubes, respectively. Chen et al. [18] designed and evaluated the performance of a PVT with cogeneration of freshwater and electricity by mixing a PV/T collector and desalination system and found that the combined system had converted around 70% of the solar irradiation to useful energy.

Barbu et al. [19] investigated and studied some limitations on the thermal and electrical performance of a hybrid PVT collector under different climatic conditions in Strasbourg and Bucharest during the summertime. The findings demonstrated that, in most cases, there is a compromise between the electrical and thermal power production. However, the thermal benefits of low wind speed and high insulation outweigh the drop in electrical-power production. Maleki et al. [20] studied the effect of coolant inlet fluid temperature on the thermoelectric behavior of a PVT collector under the climate conditions of Tehran, Iran. There were used some methods to evaluate the temperature of inlet water. The impact of the real tap-water temperature flowing through the collector on the energy-conversion efficiency was investigated, and the yearly results showed up to 4.02% higher power output compared to the models that estimate the inlet water temperature, depending on the average climate conditions. Hooshmandzade et al. [21] studied experimentally the effect of pure water, single nanofluids (0.1, 0.3, and 0.5 wt%), and mixed nanofluids ($SiO_2$-$Al_2O_3$) as coolants in PVT systems. The results showed that mixed nanofluids were the best in terms of enhancing the total efficiency of both indoor and outdoor systems. Hasan et al. [22] investigated the impact of jet of water on the thermofluidic characteristics of PVT solar collectors. Different ranges of 10,000 to 30,000 of the Reynolds number, diameter of nozzle, and jet-to-PV high (h) had been examined. The results showed an important thermal enhancement due to applying the water jet-impingement to PV-cell cooling. Many other studies were carried out to present the impact of heat-transfer enhancement on the performance of different solar systems in different engineering application [23–31].

According to the preceding literature review, several attempts have been made by researchers to assess the applicability of PVT systems in a variety of climate conditions. However, verification of the applicability of PVT systems in hot subtropical climates, such as that in the southern region of Iraq, is limited. This zone features a high solar irradiance intensity of around 2310 kWh/m$^2$/year [32]. Therefore, this research is planned to evaluate and compute the enhancement rates in the electrical efficiency, the thermal efficiency, the overall efficiency, and the output electrical power that result from using that water-based cooling method for the PV thermal management. In the course of this research, both

the influence of varying levels of solar irradiation and the pace at which cooling fluid (water) flows through the PVT solar system were parametrically compared and analyzed under real-world conditions. The enhancement that results from utilizing the water-cooled channel as opposed to a PV module (without cooling) was examined experimentally under the climatic conditions (wind speed, ambient atmosphere temperature, solar irradiation, and humidity) of the southern region of Iraq during the summer season. Experiments are carried out at the city of Al-Amarah, which is located in the south of Iraq, throughout the month of June between the hours of 9:00 A.M. and 4:00 P.M.

## 2. Experimental Setups

### 2.1. System Description

An experimental rig of a PVT solar assembly was constructed in order to assess the entire performance when a water-cooled channel is attached to the PV backside for temperature-regulation purposes. The performance was evaluated in comparison to the electrical and thermal performance of a PV module that was installed concurrently with the PVT solar system under the weather conditions of Al-Amarah, Iraq. The data from the experiments conducted on both systems were gathered at almost the same times of the day. The overall layout of the PVT solar system is shown in the schematic illustrated in Figure 1. Real-world images of the experimental configuration of the PVT solar system are shown in Figure 2A,B. A metal sheet is fastened beneath the PV module in order to create a conduit for cooling water to circulate. This helps to remove heat from the photovoltaic solar cells, which in turn improves the electrical efficiency of the system overall. A black rubber sheet was adhered to the back of the heat sink using thermal glue to prevent heat loss from the PVT system. To recirculate the cooling fluid (water), a water pump is used to regulate the mass flow rate at nominal values of 4, 8, 12, and 16 L/min. A flow meter was used to get an accurate reading of the flow rate. Water is inexpensive, non-toxic, and available over most of the Earth's surface. Water was used as a cooling fluid because it features a relatively higher specific heat capacity among commonly available liquids at room temperature and atmospheric pressure, allowing efficient thermal management over lower-sized heat sinks with low rates of mass transfer [1].

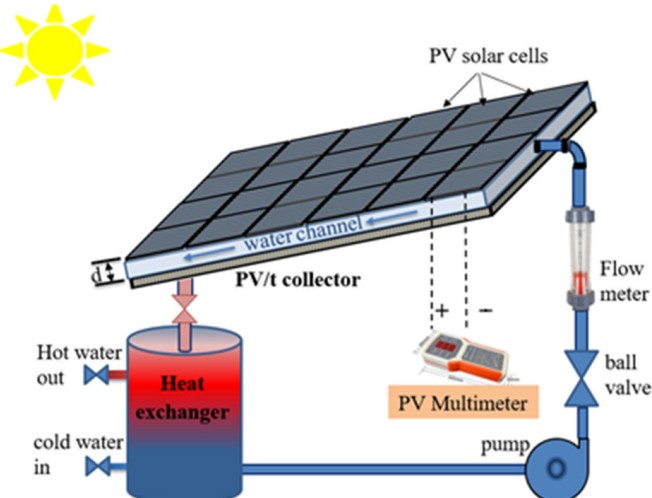

**Figure 1.** Schematic diagram of the PVT solar system.

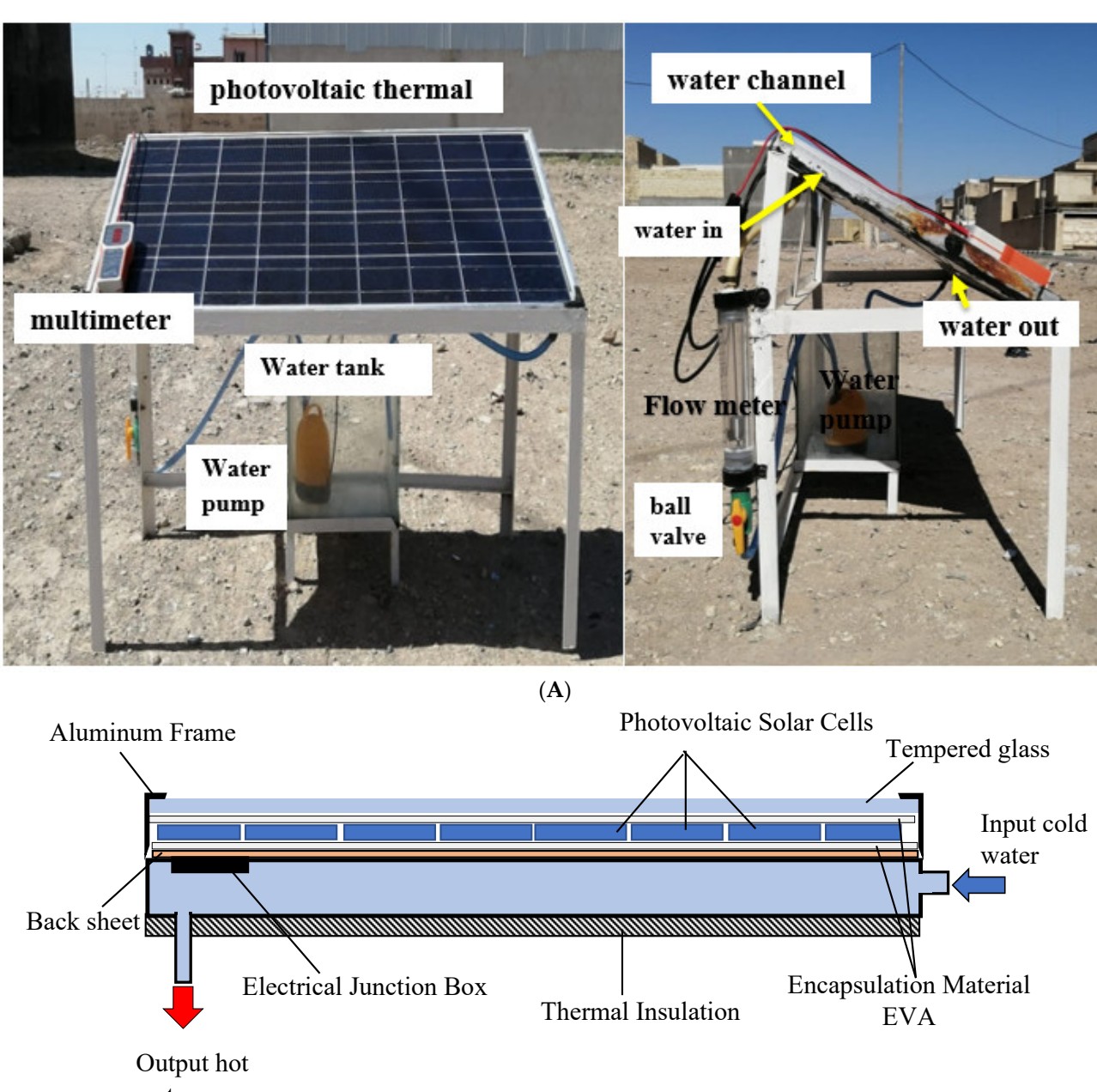

**Figure 2.** (**A**). The experimental setup of the PVT solar system. (**B**). Schematic of the PVT cross section with internal components.

### 2.2. Methods and Instruments

A Data Acquisition (DAQ) system was specifically designed and constructed to accommodate its employment in the field-testing (outdoors). It mainly consists of the following components: (1) computer and related types of software, (2) thermocouples, (3) pyranometer, (4) anemometer, and (4) multimeter PV-panel tester (MPPT). The DAQ process gets started after the AUTO MPPT test is initiated, so that the instrument immediately activates the digital display and adjusts the test interval time corresponding to the power value that is being measured. Open-circuit voltage detection test is activated when the VOC test is activated, so the voltage that the solar panel is producing will be displayed in real time. Over-voltage protection, over-temperature protection, over-current protection, and over-rated power protection are all included in the multi-protection feature set.

Tables 1 and 2 provide data of the PV module and PV/T collector specs, respectively. The performance of the PVT solar system is evaluated at several levels of solar irradiation of 500, 600, 700, 800, 900, and 960 w/m$^2$. The temperature of the plate, the water intake, and the outflow have all been measured using a set of thermocouples placed in the proper positions. The temperature of the mean PV cells and the glass has been measured using a set of thermocouples that have been permanently installed. The readings on the PVT solar system were carried out in real-world outdoor conditions on days with a clear sky on 22 June 2021. Every day, from nine in the morning until four in the afternoon, measurements were taken of the wind speed, the reading on the manometer, the outflow temperature, the input temperature, and the amount of insolation.

**Table 1.** The PV-module typical electrical characteristics under standard operating conditions.

| | | |
|---|---|---|
| Maximum power | (Pmax) | 100 W $\pm$ 2%W |
| Maximum power voltage | (Vpm) | 18.6 V |
| Maximum power current | (Ipm) | 5.38 A |
| Open-circuit voltage | (Voc) | 22.8 V |
| Short-circuit current | (Isc) | 5.76 A |
| Nominal voltage | (Voc) | 12 V |
| Tolerance | | $\pm$5% |
| Temperature coefficient of Voc | | $-0.36\%$/K |
| Temperature coefficient of Pm | | $-0.46\%$/K |
| Temperature coefficient of Isc | | $-0.05\%$/K |
| Nom. operation cell temperature | | 48 °C $\pm$ 2 °C |
| PV module electrical efficiency | | 14.8% |
| Maximum series fuse rating | | 12 A |
| Maximum system voltage | | 600 V |

**Table 2.** Specifications of the Photovoltaic–thermal solar system.

| Condition | Symbol | Value | Condition | Symbol | Value |
|---|---|---|---|---|---|
| Ambient temperature | $T_a$ | 297 K | Tilt (slope) | ° | 15 |
| Collector width | b | 0.505 m | Fluid mass flow rate | | 4–16 L/min |
| Collector parameter | P | 3.3 m | PV transmittance | | 0.88 |
| Collector area | Ac | 0.85 m$^2$ | Back-insulation conductivity | $k_b$ | 0.045 W/m$^2$ K |
| Number of glass cover | N | 1 | Water-specific heat | Cp | 4180 J/kg K |
| Emittance of glass | $\varepsilon_g$ | 0.88 | Heat-transfer coefficient | $h_{ca}$ | 45 W/m$^2$ K |
| Emittance of plate | $\varepsilon_p$ | 0.95 | PV absorptance | $\alpha$ | 0.95 |

*2.3. Theoritical Analysis*

Two distinct configurations of the PVT system are evaluated in the present analysis. One with cooling via water channel (w cooling) and one without (*w/o* cooling). For each configuration, the electrical conversion efficiency, thermal efficiency, and overall efficiency are estimated and compared. Basically, the overall efficiency of a PVT system is related to the electrical and thermal efficiencies as:

$$\eta_{PVT} = \eta_T + \eta_e \tag{1}$$

Since the effectiveness of the PVT system is depending on various operating parameters, in the current analysis, the functionality of the PVT system was evaluated at various mass flow rates. The PVT system was assumed to be a single-glazing flat-plate solar collector, so that the thermal efficiency could be estimated using Hottel-Whillier equations.

$$\eta_{th} = Q_u/I \tag{2}$$

The overall heat gain collected by PVT solar system evaluated, based on inlet and outlet temperature, mass flow rate as:

$$Q_u = \dot{m}C_p(T_o - T_i)$$

The electrical efficiency of the PVT solar system evaluated based on the temperature of PV solar module:

$$\eta_{PV} = \eta_r(1 - \gamma(T_c - T_r)) \tag{3}$$

where $\eta_r$, is the reference efficiency of the *PV* module ($\eta_r$ = 0.12), $\gamma$ is a temperature coefficient ($\gamma$ = 0.0045 °C), $T_c$ is the cell temperature, and $T_r$ is the reference temperature.

### 2.4. Uncertainty Analysis

It is essential to ascertain the degree of uncertainty present in the measured quantities of experiments. Applying Gauss's rule of propagation allowed for the estimation of the experimental uncertainties. The output R as a function of the reference parameters $x_1$, $x_2$, $x_3$, ... , $x_n$, while $w_1$, $w_2$, $w_3$, ... , $w_n$ are meant to reflect the uncertainties in the dependent parameter according to the equation below.

$$W_R = \left[\left(\frac{\partial R}{\partial x_1}w_1\right)^2 + \left(\frac{\partial R}{\partial x_2}w_2\right)^2 + ... + \left(\frac{\partial R}{\partial x_n}w_n\right)^2\right]^{1/2} \tag{4}$$

The experimental uncertainties associated with the reference parameters measured in the experiments are summarized in Table 3.

**Table 3.** Uncertainties associated with the individual elements of the PVT solar system.

| Equipment | Parameter | Experimental Uncertainty |
|---|---|---|
| Pyranometer | Solar irradiation | $\pm 3$ w/m$^2$ |
| Pressure gauges | Pressure | $\pm 3.2\%$ |
| Thermocouples | Inlet temperature | $\pm 1.09$ °C |
| Thermocouples | Outlet temperature | $\pm 1.09$ °C |
| Thermocouples | Ambient temperature | $\pm 1.09$ °C |
| Thermocouples | Glass temperature | $\pm 1.09$ °C |
| Thermocouples | PV cell temperature | $\pm 1.09$ °C |
| Multimeter | Voltage | $\pm 0.05\%$ |
| Multimeter | Current | $\pm 0.003\%$ |
| Flowmeter | Mass flow rate | $\pm 2$ L/min |

### 2.5. Experimental Error

During the course of the investigation, an error rate for the performance was determined and computed. As can be seen in Figure 3, the margin of error for the pressure reading is about $\pm 3.2\%$, but the error margin for the temperature reading is approximately $\pm 1.09$ °C. This inaccuracy was due to the equipment that was in the measurement. The temperature reading had a more accurate error than the pressure reading. The errors of around 0.05 and 0.003% were expected with a multimeter when measuring voltage and current, respectively.

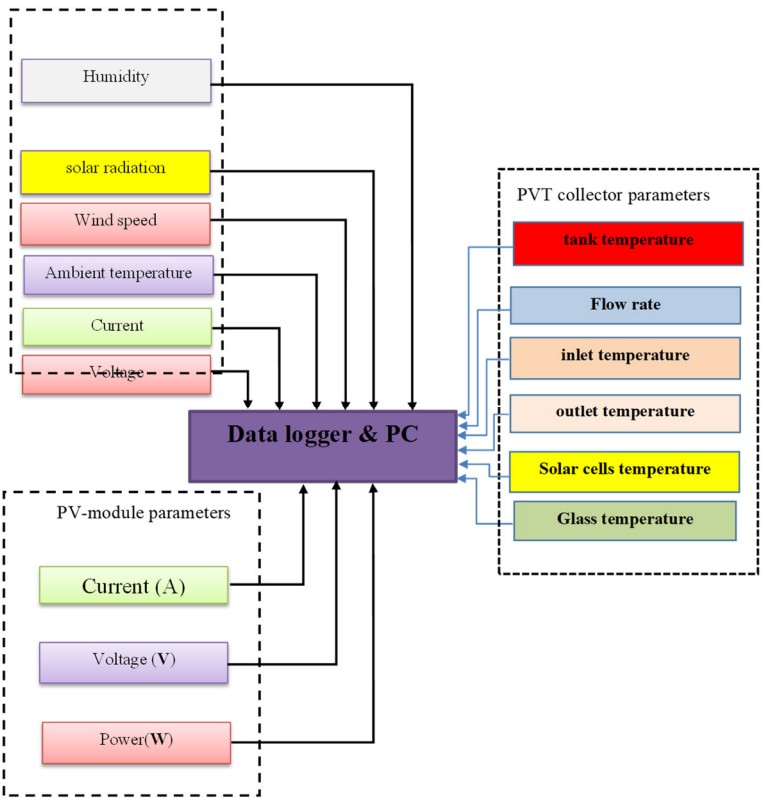

**Figure 3.** The measuring equipment flowchart for the PVT solar system.

## 3. Results and Discussion

The thermoelectrical performance of the water-cooled PV/T system was experimentally tested between 9 a.m. and 4 p.m. during the warmest month of the year, June, in the city of Al-Amarah (31.849 N, 47.145 E), in the south of Iraq. The hourly variations of climatic parameters including solar irradiation and ambient temperature on 22 June 2021 are represented in Figure 4. The maximum values were recorded at around 1 p.m., with readings of 47.5 °C and 960 W/m$^2$ for the ambient air temperature and solar irradiation intensity, respectively.

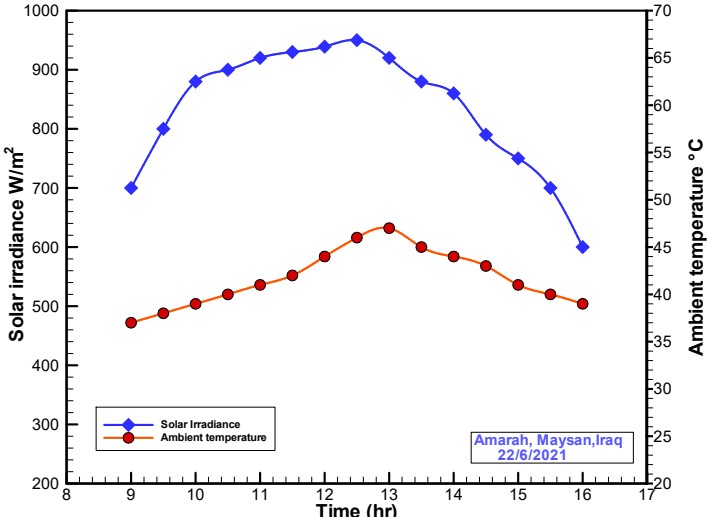

**Figure 4.** Hourly variations of ambient air temperature and solar irradiation intensity on 22 June 2021.

As a function of solar irradiation, Figure 5 depicts the variation of the mean PV-cell temperature for both PVT systems with and without cooling. The results reveal that the mean cell temperature rises as solar irradiation rises. It was also determined that the mean cell temperature of the PV module reaches 83 °C before dropping to 45 °C for the PVT solar system, owing to cooling by water at a solar irradiation of 960 w/m². The higher intensity of solar irradiation boosts the PV temperature higher, therefore, higher flow rates of water are required to regulate the temperature difference between the cooling fluid and the PV cell. Meanwhile, recycling water increases the real size of the water flow throughout the system, which subsequently enhances heat removal and reduces the unwelcome PV overheating.

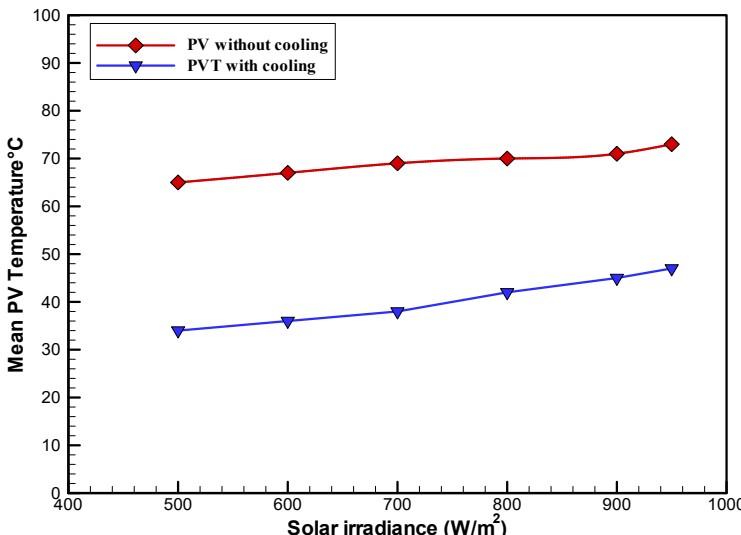

**Figure 5.** Variations of mean PV temperature at different levels of solar irradiation.

Figure 6 shows the electrical efficiency for a PVT solar system with cooling and a PV module without cooling under different solar-irradiation intensities during the day. Parameter analyses are conducted on water low rate ($\dot{m}$ = 16 L/min), solar irradiation intensities (Ipv = 500–960 W/m²), and water inlet temperature ($T_{in}$ = 25 °C). Data from this figure clearly demonstrate that the electrical efficiency decreases as solar irradiation increases because of the mean PV temperature increasing. It is also noted that the electrical efficiency of the PVT solar system is higher due to cooling by water, which in turn reduces the mean PV temperature and substantially boosts the PV power yield. On average, the incorporation of a water-cooling channel can increase the average electricity efficiency of the PVT system from 12% to 14% for all solar-irradiation intensities under consideration.

Figure 7 depicts the variation in electrical efficiency and mean PV temperature for PV modules with and without cooling as a function of daytime solar irradiation. From the data shown in Figure 7a, it is evident that electrical efficiency declines as solar irradiation increases. It is also evident that the mean PV module temperature increases from 65 °C to 75 °C when solar irradiation increases from 500 to 960 W/m² in the no-cooling configuration. The findings also reveal that solar irradiation has a significant impact on the average PV temperature and, consequently, the electrical efficiency of a PVT solar system, when water is employed as the cooling fluid. It is evident from Figure 7b that the mean PV temperature and the heat absorbed rise with increasing solar radiation throughout the day and reach their peak at midday. It was also discovered that the electrical efficiency of the PVT solar system decreases when solar irradiation increases, due to the temperature rise of the PVT solar system.

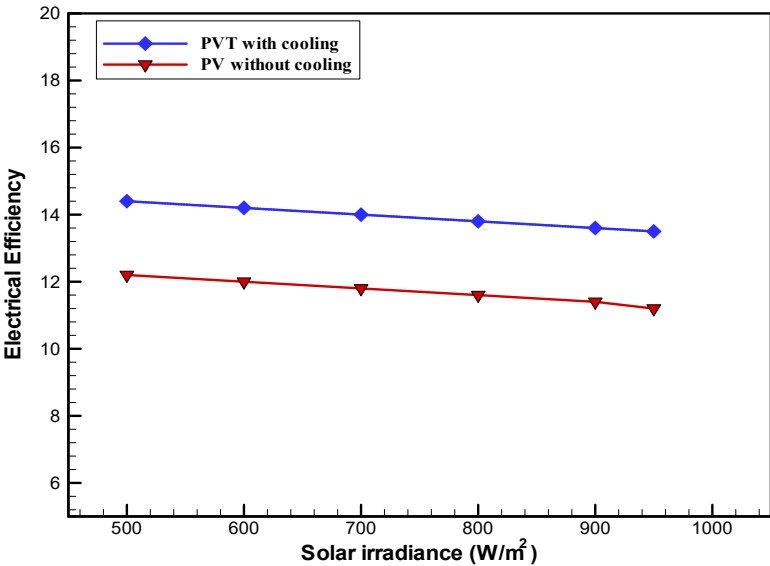

**Figure 6.** Variations of electrical efficiency at different solar radiation for photovoltaic solar system with and without cooling.

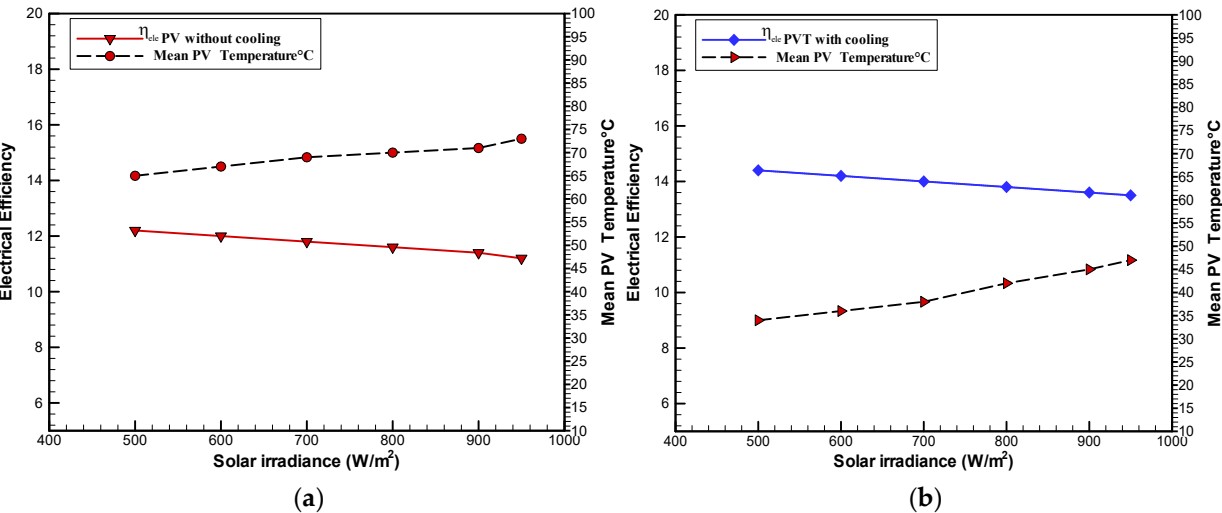

(**a**)                (**b**)

**Figure 7.** Variations of electrical efficiency and mean PV temperature at different solar radiation for (**a**) the PVT system without cooling and (**b**) the PVT system with cooling.

The mean PV temperature of the solar system increases in response to any increases in the amount of incident solar irradiation, as shown in Figure 8a,b. In Figure 8a, the mean temperature of the PV cells rises from 35.7 to 47 °C when the solar irradiation rises from 500 to 960 W/m$^2$, when water is used as the coolant fluid. It should also be mentioned that such a rise in solar irradiation results in an increase in the amount of electrical output from the PVT system. For example, increasing the solar irradiation during the daytime from 500 to 960 W/m$^2$ led to an increase in the output power from 60 to 75 W, as shown in Figure 8b, which also demonstrates that the power output can be significantly increased. When there is a greater amount of solar irradiation, the mean PV temperature of the PV module increases. The same figure demonstrates that an increase in solar irradiation from 500 to 960 W/m$^2$ results in an increase in mean PV temperature from 65 to 73 °C, when no cooling is used. It should also be mentioned that a rise in solar irradiation results in an increase in the mean PV temperature, which negatively affect the increasing trend of electrical output. In the same figure, the mean PV temperature rises from 65 to 73 °C when increasing the solar irradiation from 500 to 960 W/m$^2$ without cooling. Figure 8b

shows that the output power can be improved from 35 to 55 W when increasing the solar irradiation from 500 to 960 W/m² during the daytime. This also implies that the output power can be substantially increased when increasing the incident solar irradiation.

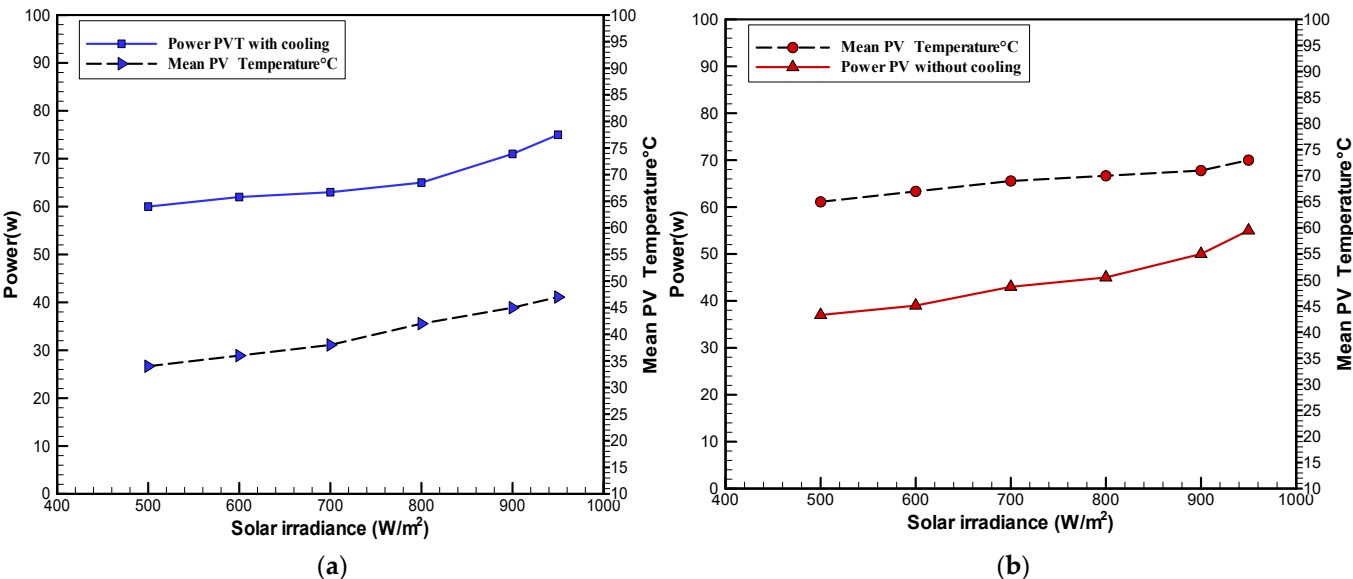

**Figure 8.** Variations of output power and mean PV temperature at different solar radiation for (**a**) the PVT system without cooling and (**b**) the PVT system with cooling.

The performance curves in Figure 9 represent the variations in output power and voltage that occur for the PVT solar system with cooling and the PV module without cooling, when subjected to the highest value considered for solar irradiation, i.e., IPV = 960 W/m². It can be seen clearly from the figure that the electrical output for the PVT system with cooling reaches 75 W, whereas the output for the PV module without cooling is only 55 W. As a result, the incorporation of a water-channel cooling system into a PVT solar power-generation system has a considerable impact on the amount of electricity generated and the electrical efficiency of the system. As previously stated, this is due to a decline in the mean temperature of the PV cells, which is caused by the removal of surplus heat from the solar cells via the cooling channel.

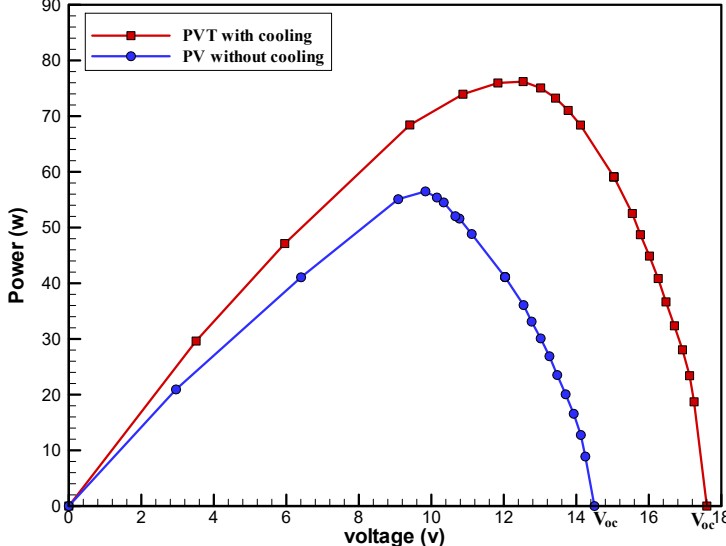

**Figure 9.** Variations of output power and voltage for PVT solar system with and without cooling.

Figure 10 shows the impact of applying different water mass flow rate on the electrical performance of PVT solar system in the middle of the day. As is expected, the electrical performance of PVT solar system is adversely affected by the water flow rate in charge. According to Figure 10, the maximum electrical efficiency of PVT solar system was observed at the water mass flow rate of 16 LPM. Therefore, it was recorded that the optimum operating temperature for the module was 45 °C at a mass flow rate of 16 LPM. It is also noted that the electrical efficiency of a PVT solar system increases when increasing the mass flow rate while decreasing the mean PV temperature. On average, the electrical efficiency can be increased from 11.7% to 14% with reducing the mean PV temperature from 73 to 45 °C. Therefore, the impact of the water flow rate on reducing the mean PV temperature and improving the electrical efficiency becomes more significant at high levels of solar irradiation, which are mostly reached in the middle of the day.

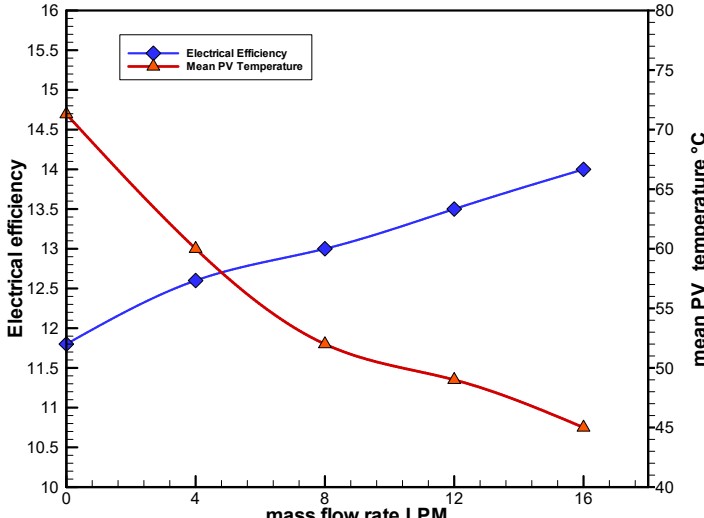

**Figure 10.** Variations of electrical efficiency and mean PV temperature at different mass flow rate for PVT solar system with cooling.

Figure 11 shows the effect of different water mass flow rates on the short circuit current of a PVT solar system in the middle of the day. As is expected, the water mass flow rate improved the short circuit current of a PVT solar system from 3.7 to 4.7 A, when reducing the mean PV temperature from 73 to 45 °C. According to Figure 11, the highest short circuit current of PVT solar system was observed at a water mass flow rate of 16 LPM. It is also noted that the short circuit current of a PVT solar system increases when increasing the mass flow rate while decreasing the mean PV temperature.

The impact of changing the mass flow rate of water on the short circuit current in a PVT solar system in the middle of the day is presented in Figure 12. The water mass flow rate enhances the open circuit voltage of a PVT solar system from 13.2 V to 16.8 V when reducing the mean PV temperature from 73 to 45 °C. According to Figure 12, the highest open circuit voltage of a PVT solar system was observed at a water mass flow rate of 16 LPM. It is also noted that the open circuit voltage of PVT solar system increases when increasing the mass flow rate while decreasing the mean PV temperature.

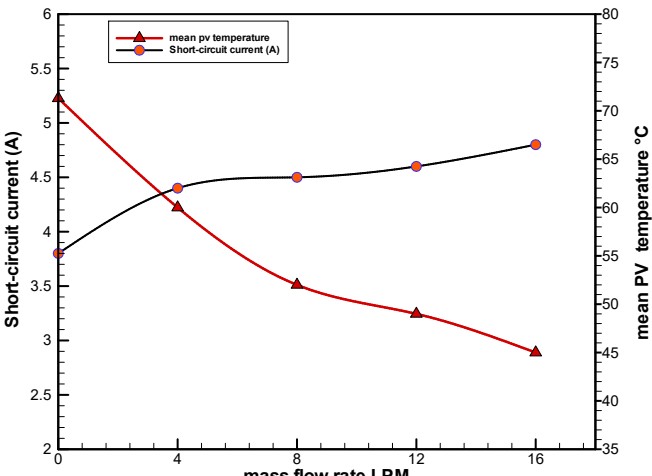

**Figure 11.** Variations of short circuit current and mean PV temperature at different mass flow rate for PVT solar system with cooling.

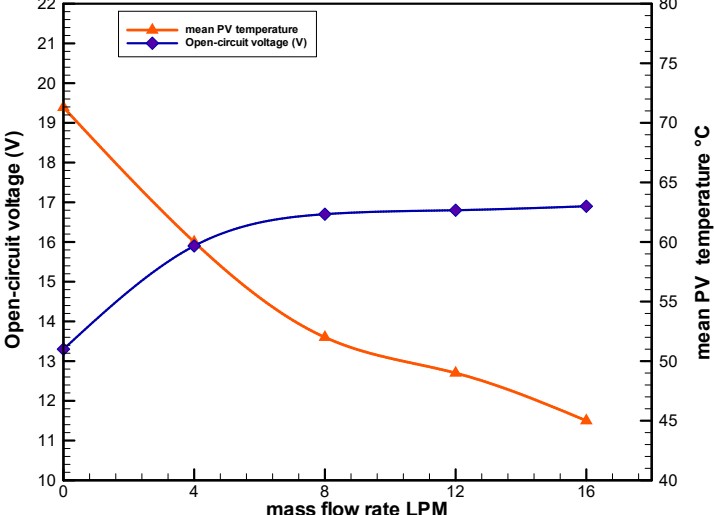

**Figure 12.** Variations of open circuit voltage and mean PV temperature at different mass flow rate for PVT solar system with cooling.

The impact of varying the mass flow rate of water on the mean PV temperature in a PVT solar system during daytime is displayed in Figure 13. From the results, it is clearly noted that the mean PV temperature decreases when increasing the mass flow rate for all of the time during the day. The highest mean PV temperature was recorded in the middle of the day for the PV module without cooling. The lowest mean PV temperature was recorded in the middle of the day for the PVT solar system with cooling with a mass flow rate of water of 16 LPM.

Figure 14 presents the variations of mean PV temperature and electrical efficiency at 16 LPM for PVT solar system during the daytime. As is expected, the electrical efficiency of a PVT solar system decreases when increasing the mean PV temperature. According to the data, the maximum electrical efficiency of a PVT solar system was observed at the lowest mean PV temperature at 9 AM and the minimum electrical efficiency at the relatively highest mean PV temperature in the middle of the day.

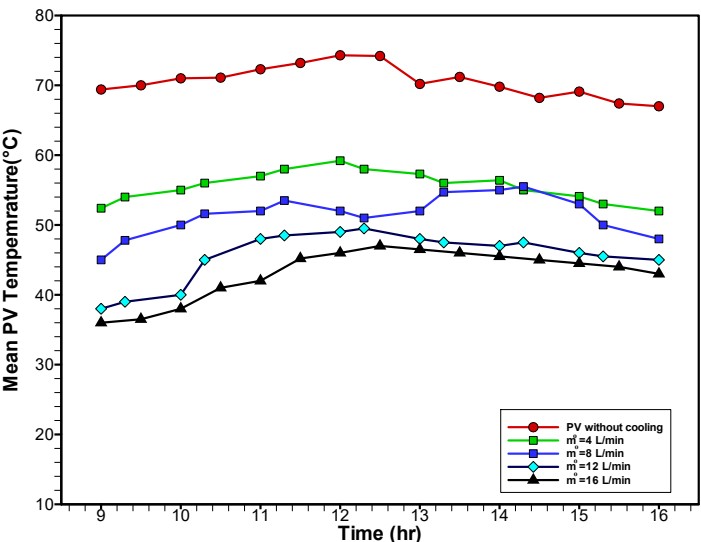

**Figure 13.** Variations of mean PV temperature at different mass flow rate for PVT solar system during daytime.

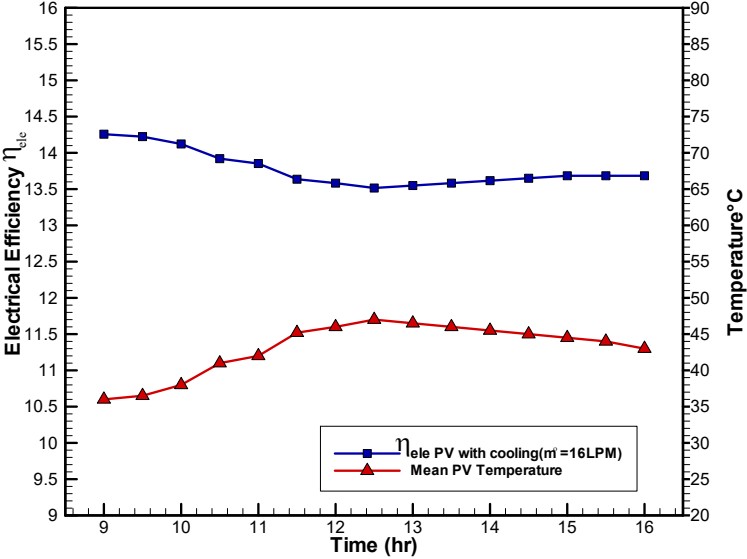

**Figure 14.** Variations of mean PV temperature and electrical efficiency at 16 LPM for PVT solar system during daytime.

## 4. Conclusions

A photovoltaic–thermal (PVT) solar system was constructed and experimentally evaluated in the climatic conditions of Al-Amarah, Iraq, during the month of June. Regarding the improved performance of the entire system due to the introduction of a water-cooled channel, the following conclusions can be drawn:

(1) The effects of varying the mass flow rate of cooling water of 4, 8, 12, and 16 L/min inside the cooling channel on the mean PV temperature during the daytime were presented, and it was evident from the obtained data that the mean PV temperature decreases as the mass flow rate increases at all times during the day.

(2) The mean PV temperature increases from 65 to 73 °C as solar irradiance increases from 500 to 960 W/m². The highest average PV temperature was observed at midday for modules without cooling. The lowest mean PV temperature was recorded at midday for a PVT solar system with water cooling at a mass flow rate of 16 LPM. The open circuit

voltage of a PVT solar system increases as the mass flow rate and mean PV temperature decrease.

(3) The mean PV temperature rises as solar irradiation rises; the mean PV temperature of a PV module reaches 83 °C and then drops to 45 °C for a PVT solar system, owing to the cooling by water at a 960 w/m$^2$ solar-irradiation level.

(4) As solar irradiation increases, the electrical efficiency declines as the mean PV temperature rises; however, the electrical efficiency of the PVT solar system is enhanced by the incorporation of a water-cooled channel.

(5) The electrical efficiency decreases as solar irradiation increases because of the mean PV temperature increasing, and the electrical efficiency of a PVT solar system is improved due to the introduction of a water-cooled channel.

(6) The higher mass flow rates of water improve the electrical efficiency of a PVT solar system from 11.7 to 14% when reducing the mean PV temperature from 73 to 45 °C.

(7) The electrical efficiency of a PVT solar system rises as the mass flow rate and average PV temperature decrease. The maximum electrical efficiency of a PVT solar system is recorded at 16 L/min of water mass flow.

(8) Adding a water-cooled channel to a PVT solar system has a substantial impact on the system's output power and electrical and thermal efficiencies, as a result of a reduction in the average temperature of the sun cells. The power output of a PVT solar system with cooling is 75 watts, whereas the power output of a PV module without cooling is 55 watts.

Finally, it would be helpful for future research directions to include investigating the effects of geometrical parameters of the cooling heat sink as well as testing different types of cooling fluids and their effects on the electrical and thermal performance of the PVT system.

**Author Contributions:** Conceptualization, H.A.H. and J.M.M.; Data curation, H.T.; Formal analysis, W.Y.; Funding acquisition, H.A.H. and A.M.A.; Investigation, J.M.M., A.M.A. and R.K.I.; Methodology, H.T.; Project administration, H.A.H. and W.Y.; Resources, J.S.S.; Supervision, R.K.I.; Writing—original draft, J.S.S.; Writing—review & editing, J.S.S., J.M.M. and H.T. All authors have read and agreed to the published version of the manuscript.

**Funding:** This research received no external funding.

**Conflicts of Interest:** The authors declare no conflict of interest.

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
