# Peer review of "Experimental Evaluation of the Thermoelectrical Performance of Photovoltaic-Thermal Systems with a Water-Cooled Heat Sink"

_sustainability, doi:10.3390/su141610231_

Round 1
Reviewer 1 Report
The presented article presents the results of a full-scale study of a solar photovoltaic thermal module. The topic of the article is relevant and may be of interest to specialists and researchers in the fields of solar energy and the optimal use of resources. The article presents interesting results of the experiments, the authors have done some work, however, as critical remarks and important recommendations, several points should be noted:
1. What is the scientific novelty of the work done, as well as the conclusions obtained by the authors?
2. More detailed theoretical calculations of the thermal and electrical modes of operation of the photovoltaic thermal module are of interest.
3. Three-dimensional modeling in the system of finite element analysis of the developed module is also interesting for comparison and agreement with the experimental results obtained.
4. As a sealing material for photovoltaic converters for solar modules in hot climates, the authors are recommended to consider in a literature review or their future studies a two-component polysiloxane compound (for example, DOI: 10.4018/978-1-7998-9152-9.ch030), which is appropriate in structures with thermal functions due to increased electrical efficiency and no degradation.
5. How is the front and rear thermal insulation of the coolant of the proposed module provided?
6. What is the optimum operating temperature for the module, based on overall efficiency? At what flow rate?
7. What are the advantages of planar photovoltaic thermal modules compared to concentrator modules (for example, DOI: 10.4018/978-1-5225-3867-7.ch004)?
8. At the beginning of the introduction, references to all sources of information used are necessary. The sources used should be indicated by numbers in square brackets as required by the publisher.
9. The authors should justify the choice of module design in more detail, since there are a significant number of designs of planar photovoltaic thermal modules (they should also be briefly described).
10. Also, the authors should add a drawing/sketch of the section of the photovoltaic thermal module with an image of its internal structure.
11. The authors should pay attention to "Temperature coefficient of Isc -0.05%/K" - how will the electric current change when the photovoltaic converters are heated? After Figures 1 and 2, conclusions should be drawn on the design and system developed. Formula 3 (lines 206-207) should be marked, all symbols used in all formulas should be deciphered and the text format should be made uniform. Table 3 is missing from the text of the work. Figure 3 should be enlarged and conclusions drawn after it. Figure 5 does not show a dot for 83°C as indicated by the authors.
12. The authors should describe in more detail the technical parameters of the measuring equipment and sensors used in Figure 3. Where is this equipment installed in Figures 1 and 2? It is advisable to add a subsection "Methods and Instruments" to the text of the work.
13. What is the cost of the installed power of the resulting module compared to a photovoltaic module and a thermal collector, and how quickly will such a module pay for itself?
14. What formulas are usually used to calculate changes in open circuit voltage and short circuit current (theoretically) and how will theoretical calculations be consistent with the experimental graphs obtained in Figures 11 and 12?
15. Of interest is the thermal and overall efficiency of the developed photovoltaic thermal module, as well as the obtained thermal parameters in the form of graphs.
16. After the results obtained, it is necessary to add a subsection "Discussions", where to the authors should substantiate and analyze in detail the data obtained, how applicable is the methodology used for other similar experiments, etc.
17. Also, the authors should add a subsection "Directions for further research", where they should indicate the planned research on the topic under consideration.
18. Where and how is it planned to implement the results of the work done by the authors?
In general, the presented article leaves a positive impression, however, it is not without serious shortcomings.
Author Response
Thanks for your constructive comments. Please find attached details of our response to your comments that are shown in black and our responses shown in red.

Reviewer 2 Report
In this work, Hasan et al. used a water-cooled heat sink in order to thermally manage the PV cells to boost the electrical output of the PVT system in the climatic conditions of the southern region of Iraq during the summertime. The problem addressed here is real and interesting. However, the topic of PVT systems is not new and there are several reports already available on this. Hence, I only recommend this work for publication in case the authors are ready to address the following concerns:
1) In the introduction section, there must be some discussion on ferro-nanofluids as they are known to be excellent heat conductors and frequently used as electronic coolants for PVT systems.
2) What is the rationale behind using a water-cooled heat sink? Advanced systems are using ferrofluids nowadays because their thermal conductivities are much higher than the water.
3) The obtained results must be compared with different ferrofluids/nanofluids already used as coolants, otherwise, we cannot evaluate the performance of the present PVT system using a water-cooled heat sink.
4) The % of errors should be mentioned clearly and if possible include the error bars in the plots.
5) The conclusion section should not be written in points.
Author Response
Thanks for your constructive comments. Please find attached details of our response to your comments that are shown in black and our responses are shown in red.

Reviewer 3 Report
In the paper entitled: " Experimental evaluation of the thermoelectrical performance of photovoltaic thermal systems with a water-cooled heat sink, the authors have conducted the water-cooled heat sink to thermally manage the PV cells in order to boost the electrical output of the PVT system. The authors have constructed a simple test rig and the results are obvious.
1- Where is a novelty of this study?
2- The authors have not considered the energy needed for pumping the water and I think it is important to consider it.
3- More importantly, in Middle east the main concern is shortage of the of the water not energy. Utilizing water for heat sinks in a solar power plant is a big problem.
Duo to lack of novelty and application without considering the energy conservation rule.
Author Response

(The authors gave the same response as above.)

Round 2
Reviewer 1 Report
The revised article looks better - the authors have made major corrections and adjustments. As recommendations, the authors should pay attention to the following: at the beginning of the introduction, the authors should add several references, on the basis of which the authors draw conclusions; authors should also justify the choice of the module design used (single glazing, etc.) and show the cross section of the module used (PVT, not PV) in the Figure.
Reviewer 2 Report
The revised manuscript can now be accepted in its present form.
Author Response
Thank you so much for your constructive comments that improved the quality of the work.
Reviewer 3 Report
The answers to the comments are not acceptable.
Author Response
Thanks for your comments. The paper has been heavily revised in several sections responding to the reviewers' comments.
Round 3
Reviewer 3 Report
The paper has not considered so many important things in sustainability of the energy. The idea is not novel nor feasible and the experimental data are not reliable.
I have clarified my comments on the previous reviews.
Author Response
Thank you for your feedback. Please note that the manuscript has been substantially revised, and the authors have responded to all of the reviewers' comments point-by-point. We appreciate the reviewer's time and work, but we need to have specific comments in order to respond point-by-point.
1-The reviewer states that our idea is not novel or feasible even though the evidence is provided in the paper on lines 151–155:
"According to the preceding literature review, several attempts have been made by researchers to assess the applicability of PVT systems in a variety of climate conditions. However, verification of the applicability of PVT systems in hot subtropical climates, such as that in the southern region of Iraq, is limited. This zone features a high solar irradiance intensity of around 2,310 kWh/m2/year"
2-The reviewer also states that our experimental data are not reliable.
Please note that we have conducted experiments under real-world conditions, provided details on all the specifications of the measuring devices, and done uncertainty analysis. All details are given in Sec. 2.2. Methods and Instruments, Sec. 2.4. Uncertainty Analysis, and Sec. 2.5. Experimental Error.